# Suprasellar Ganglioglioma Arising from the Third Ventricle Floor: A Case Report and Review of the Literature

**Shaoguang Li** †, **Yuanyuan Xiong** †, **Guowen Hu, Shigang Lv, Pingan Song, Hua Guo and Lei Wu** *

Department of Neurosurgery, The Second Affiliated Hospital of Nanchang University, Nanchang 330006, China
* Correspondence: doctorleiming@163.com
† These authors contributed equally in this work.

**Abstract:** Gangliogliomas are uncommon intracranial tumors that include neoplastic and abnormal ganglion cells, and show positive immunohistochemical staining for GFAP and syn. This type of lesion occurs more frequently in the temporal lobe than in other areas; they are extremely rare in the suprasellar region. To the best of our knowledge, including our case, 19 cases of GGs have been found in the suprasellar region. Among them, five tumors invaded the optic nerve, nine tumors invaded the optic chiasm, one tumor invaded the optic tract, and two tumors invaded the entire optic chiasmal hypothalamic pathway. In the present study, we describe the first case of suprasellar GGs arising from the third ventricle floor that was removed through the endoscopic endonasal approach. In addition, we summarize the clinical characteristics of GGs, such as age of onset, gender distribution, MRI signs, main clinical symptoms, and treatment methods for GG cases.

**Keywords:** ganglioglioma; suprasellar region; endoscopic operation; neuroepithelial neoplasms; neuropathology

## 1. Introduction

GGs account for 0.3~1.4% of all intracranial tumors [1]. They are a type of relatively rare, well-differentiated, low-grade intracranial tumor that is common in young people and children, with a higher prevalence in men. Seizures are the most common clinical manifestation. GG is characterized by a combination of dysplastic neurons and neoplastic glial components. Histopathologically, suprasellar GGs are so rare that, to our best knowledge, only 19 cases have been described in the literature, including the case in our present study. This study is the first case to document the case of suprasellar GGs originating from the third ventricle floor. We describe the case of an asymptomatic suprasellar GG case. A middle-aged female patient underwent endoscopic resection through a trans-sphenoidal approach, and her postoperative course was stable.

## 2. Patient Information

A 40-year-old female patient complained of occasional dizziness, no other particular discomfort, and no obvious positive signs were discovered in the neurological exam that was performed. The patient had normal vision and dot-like visual field defects in the left eye. The pituitary and its target-gland hormone levels were normal. Her pituitary MRI scan showed an oval mass with a homogeneous signal behind the suprasellar pituitary stalk, of about 17 mm × 11 mm × 17 mm in size, with a low signal on the T1 image, high signal on the T2 image, and a mild forward displacement of the pituitary stalk. There was no abnormality in the shape and signal of the pituitary stalk and pituitary itself. The MRI showed that the lesion in the suprasellar area was slightly enhanced, while the pituitary and pituitary stalk were significantly enhanced (Figure 1). Differential diagnosis of NF1: the patient did not have a first-degree relative with NF1, café au lait patches, freckling in the axillary or groin, iris Lisch nodules, cutaneous or subcutaneous neurofibromas, or one

plexiform neurofibroma. Sphenoid wing dysplasia and the thinning of the long bone cortex were also absent. To clarify the differential diagnosis, we recommended genetic testing, but the patient declined. After explaining the purpose and risks of the operation to the patient and her family, informed consent was provided. An operation, using a nasal approach, was performed to surgically remove the tumor.

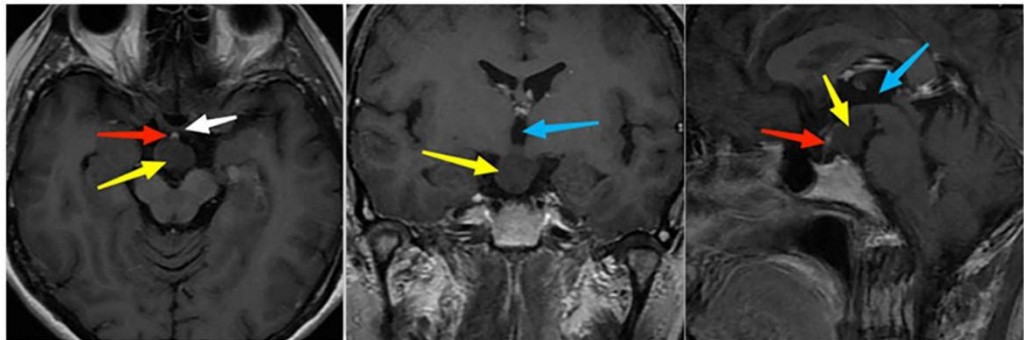

**Figure 1.** MRI images. Preoperative MRI-enhanced T1 images show a slightly strengthened tumor located in the suprasellar region, behind the optic chiasm, and below the third ventricle; the pituitary gland and pituitary stalk are significantly enhanced (yellow arrows point to the tumor, white arrow points to the optic chiasm, red arrows point to the pituitary stalk, and blue arrows point to the third ventricle).

### 3. Operative Note

The tuberculum sella was removed under endoscopy, and the dura mater was then opened. The tumor behind the optic nerve and the pituitary stalk could be observed after dissecting the arachnoid membrane (Figure 2A,B). Using the endoscope, the tumor was found to be located at the center of the back of the pituitary stalk; the pituitary stalk was pushed to one side and the arachnoid membrane around the tumor was separated. During the operation, it was observed that the tumor originated from the third ventricle floor between the pituitary stalk and the mammillary body. Unlike the third ventricle floor, the lesion retained clear boundaries and presented no adhesion to the surrounding structure. The tumor was gray–white, tough in texture, with only a few tiny vessels on the surface and no calcifications (Figure 2A,B; purple arrow). The origin of the tumor is poorly demarcated and pale yellow in the figure. After the tumor was removed, the third ventricle was also opened (Figure 2C; blue arrow). Artificial dura mater and the vascularized pedicle nasoseptal flap were used to repair the skull base defect, and routine hemostasis was performed by the expansion of a polymer sponge.

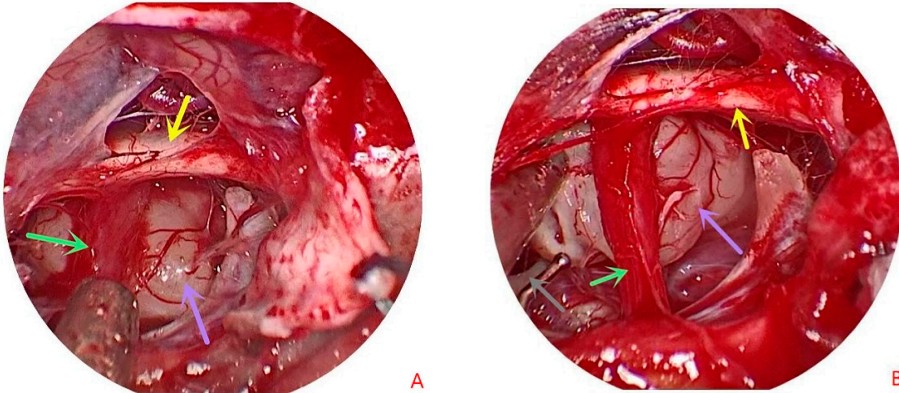

**Figure 2.** *Cont.*

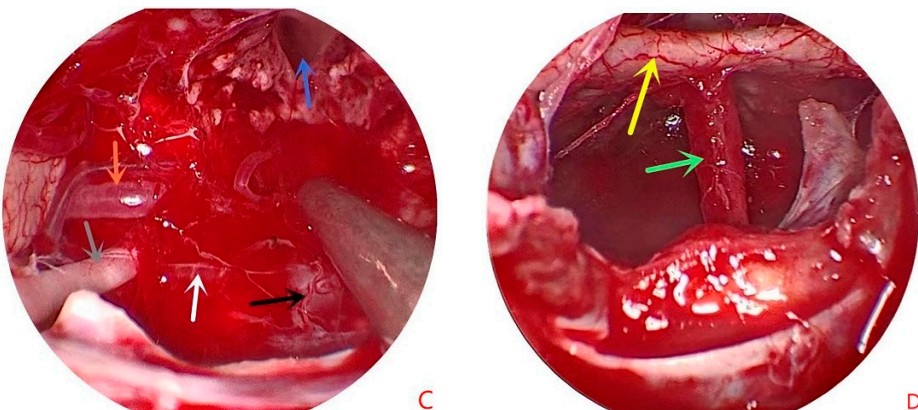

**Figure 2.** Endoscopic images during surgery. (**A**,**B**) Opening the dura mater at the bottom of the sella, a gray–white tumor (purple arrows) located behind the pituitary stalk (green arrows), and below the optic chiasm (yellow arrows). (**C**,**D**) Following the removal of the tumor, the third ventricle (blue arrow), oculomotor nerve (gray arrow), posterior cerebral artery (orange arrow), and basilar artery (black arrow) can be observed.

## 4. Histopathology

The tumor is gray–white, gross in appearance, and firm in texture. Microscopically, gliosis was observed with an increased density and enlarged nuclei. The hematoxylin and eosin-stained section presents rare dysmorphic ganglion cells (Figure 3A; yellow arrow). No obvious mitotic image can be observed, and immature neurons are randomly scattered or aggregated in a nest shape (Figure 3A; green arrow). Immunohistochemical staining shows that CD56, GFAP, S100, Syn, and Neun (partly) are positive in the tumor cells, and CK, EMA, CgA, IDH1, P53, and CD34 are negative (Figure 3B–D). The Ki-67 index is about 1%. The histopathological diagnosis was ganglioglioma (WHO grade I).

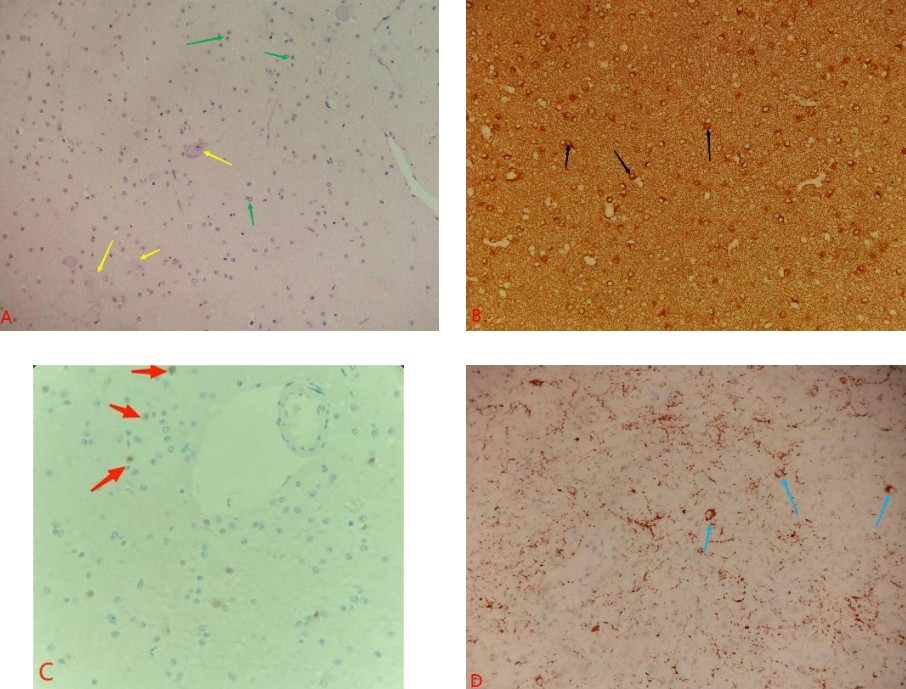

**Figure 3.** The pathological staining of tumors. (**A**) Hematoxylin and eosin-stained section showing a large number of astrocytes (green arrows) and ganglion cells (yellow arrows) with large nucleoli were scattered. (**B**) The cytoplasm of glial cells shows positive staining for GFAP (black arrows). (**C**) A few NEUN-positive nuclei are visible (red arrows). (**D**) Syn staining reveals the positive cytoplasmic staining of neoplastic neurons (blue arrows).

## 5. Postoperative Course

Following the operation, the neurological condition and endocrine function of the patient remained normal. There was no serious surgical complication, electrolyte disturbance, or cerebrospinal fluid leakage. Postoperative magnetic resonance imaging showed that the tumor was completely removed, and the pituitary gland, pituitary stalk, and optic chiasm were structurally intact in the imaging. In addition, the margin of the operation area was enhanced postoperatively—we considered that this was a sign of postoperative capillary reactional enhancement; no other obvious abnormality in the image was observed. All these results suggest that the tumor was completely removed. A pathological examination revealed that the tumor was composed of neoplastic ganglion cells and low-grade astrocytic cells. The patient's condition was stable following the operation and she was discharged with no complications. One month after the surgery, the patient received further radiotherapy treatment in the oncology department of our hospital, in consideration of the 2018 Guidelines for the Diagnosis and Treatment of Brain Glioma, which determine that, for patients aged $\geq$ 40 years or patients with an incomplete tumor resection, radiotherapy or chemotherapy is recommended. Moreover, the 2020 NCCN Central Nervous System Clinical Practice Guidelines highlight that elderly patients should receive early radiotherapy following surgery. We informed the patient and her family of this, and they expressed their desire to receive radiation therapy [2,3].

## 6. Review of Ganglioglioma Cases

We conducted an English-language search of the PubMed and Embase databases, using "ganglioglioma" as the search term; including our case, we identified 19 patients with a diagnosis of suprasellar GGs (Table 1). The average presentation age was 20.95 $\pm$ 3.42 years; the range was 6–52 years of age. Most of the patients were male (12/19). The symptoms of these patients were mainly visual dysfunction (15/19), some patients suffered from obstructive hydrocephalus (4/19) due to the compression of the third ventricle and required emergency external ventricular drainage or a ventriculoperitoneal shunt, and we discovered that some patients had rare symptoms, such as middle cerebral artery cerebral infarction, limb hemiplegia, diabetes insipidus, and abnormal weight gain, as well as symptoms related to hyperprolactinemia [4–7].

When referring to the pre-operative MRI findings of these patients, most (12/19) patients had cystic and solid tumors, and nearly half (11/19) of the tumors could be enhanced, and the degree of enhancement was uneven.

In the previous reports, biopsy was chosen in some cases (9/19), which may have been due to surgical difficulties. Six patients (6/19) received radiotherapy following surgery, and two patients (2/19) received chemotherapy [4–21]. Unfortunately, the data to assess the effect and prognosis of adjuvant therapy were unavailable.

**Table 1.** Reported cases of suprasellar ganglioglioma.

| Publications | Age, Gender | Signs and Symptoms | Symptom Duration | Location | MRI Signs | Treatment Solutions | Preoperative Diagnosis |
|---|---|---|---|---|---|---|---|
| Khoi D. Nguyen et al. [8] | 25, M | Headache, agitation, mental status decline, obstructive hydrocephalus | Acute onset | Sellar/suprasellar region | Heterogeneously enhanced 3.6 cm × 4.2 cm cystic and solid mass | STR | Craniopharyngioma |
| Shanop Shuangshoti et al. [7] | 21, F | Hyperprolactinemia, decreased vision | 2 Y 6 Mon | Suprasellar and optic chiasm | Heterogeneously enhanced 3.0 cm × 2.8 cm cystic and solid mass | NTR | Pilocytic astrocytoma |

**Table 1.** *Cont.*

| Publications | Age, Gender | Signs and Symptoms | Symptom Duration | Location | MRI Signs | Treatment Solutions | Preoperative Diagnosis |
|---|---|---|---|---|---|---|---|
| Laura Pastor et al. [5] | 9, M | Headache and epistaxis, lefthemiparesis and gait impairment (middle cerebral arteryinfarction) | 2 Mon | Suprasellar | Heterogeneously enhanced 2.2 cm × 2.5 cm cystic and solid mass | Biopsy | Germ-cell neoplasm, hypothalamic glioma |
| Ruchika Gupta et al. [6] | 7, M | Abnormal weight gain, diabetes insipidus, decreased vision, obstructive hydrocephalus | 18 Mon | Suprasellar | 4.2 cm × 4.5 cm cystic and solid mass | Biopsy + R | Germinoma |
| Rakesh Jalali et al. [18] | 7, F | Decreased vision | 6 Mon | Suprasellar and left parasellar regions | Enhanced cystic and solid mass | Biopsy + R | Optic pathway glioma, germ-cell neoplasm |
| Ishita Pant et al. [19] | 8, F | Decreased vision, headaches | 5 Y | Suprasellar, third ventricle | 7 × 6.5 cm cystic and solid mass | NTR | Chiasmatic glioma |
| John D. Rolston et al. [20] | 49, M | Developed headaches, decreased vision | Several months | Suprasellar | Enhanced 2 × 1.2 cm cystic and solid mass | Biopsy | NR |
| Ramazan Albayrak et al. [21] | 52, F | Decreased vision, headaches | Several months | Right optic tract | Diffuse contrast enhancement in right optic tract mass | Partial resection | Optic tract glioma |
| Jonathan Chilton et al. [9] | 33, M | Decreased vision | NR | Optic chiasm, optic nerve | Solid mass | Biopsy | Optic pathway glioma |
| Donald J. Bergin et al. [10] | 13, M | Proptosis, decreased vision, optic atrophy | NR | Optic nerve | 3.0 × 1.8 cm left retrobulbar optic nerve | NR | Optic pathway glioma |
| Kazuhiko Sugiyama et al. [4] | 12, F | Decreased vision, gradual hemiplegia of left limb | 4 Y | Optic nerve | Cystic and solid mass | Biopsy | Optic pathway glioma |
| Yoshito Sugita et al. [11] | 26, M | Decreased vision, headache, obstructive hydrocephalus | 1 Mon | Optic chiasm, suprasellar | Cystic and solid mass | Partially resection + V-P shunt + R | NR |
| Burak Karaaslan et al. [12] | 12, F | Decreased vision | 6 Mon | Suprasellar, optic nerve | Heterogeneously enhanced 3.0 cm × 3.0 cm cystic and solid mass | Biopsy | NR |
| G. V. Vajramani et al. [13] | 18, M | Decreased vision | 1 Y | Optic nerve, optic tract, geniculate | Heterogeneously enhanced cystic and solid mass | Biopsy + C | NR |
| G. T. Liu et al. [14] | 6, M | Headache, vomiting, decreased vision | NR | NR | NR | Biopsy + R | NR |
| William Y. Lu et al. [15] | 38, M | Decreased vision | 6 Mon | Optic nerve | Enhanced 4.8 × 0.9 cm mass | Surgery | NR |
| Pietro Spennato et al. [16] | 6, M | Decreased vision | NR | Sellar, suprasellar, third ventricle | Cystic and solid mass | Surgery + V-P shunt + R + C | NR |
| Bashar Abuzayed et al. [17] | 16, F | Decreased vision | 15 Y | Hypothalamus, optic chiasm | Uniformly enhanced solid mass | Surgery | NR |
| ShaoGuang Li et al. | 40, F | Dizziness, visual field defect | NR | Suprasellar | Enhanced 1.7 × 1.1 cm cystic mass | GTR + R | Craniopharyngioma |

R: radiation therapy; C: chemotherapy; GTR: gross total resection; NTR: near-total resection; STR: subtotal resection; NR: not reported; M: male; F: female; Y: years; Mon: months; V–P shunt: ventriculoperitoneal shunt.

In addition, we also searched for GGs in the sellar region and ventricle. We discovered 5 cases of GGs in the hypophysial fossa and 15 cases in the ventricle. Among the five

cases, with the exception of one case accidentally discovered via autopsy, four patients had hormonal disorders in clinical manifestations. Among the four patients, two cases had elevated growth hormone levels, one case had elevated prolactin levels, and one case had elevated antidiuretic hormone levels. All cases received surgical treatment, and the operation was successful [22–25]. Among the 15 patients with ventricular GGs, 9 tumors (9/15) were located in the lateral ventricle, 5 tumors (5/15) in the third ventricle, and 1 tumor (1/15) in the fourth ventricle. Among these cases, eight patients (8/15) had dilated ventricles and seven patients (7/15) had decreased visual acuity, in which two patients (2/7) recovered in their visual acuity in a short period of time following surgery, two patients (2/15) had tumor hemorrhages, one patient (1/15) had intramedullary dissemination and metastasis, and three patients (3/15) had hydrocephalus that could not be relieved following surgery, which was subsequently improved by the insertion of a ventriculoperitoneal shunt [26–35].

From the literature review, we determine that the differences between optic pathway tumors and third ventricle tumors include the following concepts: in terms of clinical manifestations, ventricular tumors are generally characterized by increased intracranial pressure, such as a severe headache, projectile vomiting, and papilledema. Visual path tumors often present changes in visual acuity or the visual field, such as obvious monocular or binocular vision loss or loss of visual field. Optic path tumors may present a thickening of the nerves, and we believe that warrants the image identification of tumors in the above two positions.

## 7. Discussion

The term ganglioglioma, first referred to by Perkins in 1926 and popularized by Courville in 1930, is used to describe a type of central nervous system tumor containing astrocytic and neuronal components, the pathogenesis of which is unknown. According to a statistical analysis of 326 ganglioglioma cases, the mean age of onset in the 279 ganglioglioma grade I tumors was $22.1 \pm 11.2$ years old, 30 grade II ganglioglioma cases had a similar age distribution, and in 17 anaplastic gangliogliomas (WHO grade III), the mean age was approximately $35 \pm 14.5$ years old [1]. The main symptom of gangliogliomas in the temporal lobe is refractory epilepsy [36], while in other parts, it is generally caused by focal nerve compression or increased intracranial pressure [4,9,10,26,34]. Infratentorial GGs may present with cerebellar signs, cranial nerve deficits, or, rarely, increased ICP [37]. Intracranial GGs are most commonly found in the temporal lobe, but rarely in the pituitary gland, pituitary stalk, optic pathway, hypothalamus, ventricle, and spinal cord. Most GGs grow slowly and have clear boundaries, cystic components, and calcifications. GGs are generally classified as WHO grade Ia, and some GGs have anaplastic features and are considered to be grade III ganglioglioma (anaplastic GG). If the tumor contains mitosis, necrosis, and microvascular proliferation, it is generally a WHO grade IV ganglioglioma with glioblastoma changes [38]. Histopathologically, the admixture of neoplastic cells and abnormal ganglion cells is the diagnostic marker; ganglion cells can be identified by the presence of Nissl substance and large nucleoli [39]. It is known that ganglion cells or neurons in GGs are dysplastic due to cellular disorganization, subcortical localization, aggregation, and the accumulation of giant cells and Nissl substance around the membrane [1]. Immunohistochemistry staining for glial fibrillary acidic protein (GFAP), S-100 protein, and neuronal markers (Neun, Syn) helps to identify neuronal and glial cell populations. The semi-quantitative assessment of the Ki-67 labeling index helps characterize the biological behavior of the tumors [39]. A study found that the expression of CD34 in GGs may be related to the location of the tumor, and the staining in temporal lobe lesions was mostly positive for CD34, while in other rare sites it was mostly negative [1]. Generally, low-grade GGs have a low Ki-67 index (<1%), without *p53* mutations. Tumors with high levels of *p53* mutations and a Ki-67 index generally have anaplastic features, which may suggest high-grade features [40].

On imaging, GGs can be divided into cystic tumors (more than a 90% cystic component), cystic solid tumors (cystic component 10–90%), and solid tumors (cystic component less than 10%), of which cystic tumors account for about 44–80% [41]. In the MRI images, GGs generally present an iso-intense signal or a slightly hypo-intense signal on T1-weighted images, and a hyperintense signa in T2-weighted images. Generally, there is no peritumoral edema. When an edema occurs, it generally indicates a malignant lesion [42]. About half of GGs show an enhancement in MRI images after gadopentetate dimeglumine injection, and the degree of enhancement is inhomogeneous. It usually appears as a low-density lesion on the CT, with varying degrees of enhancement or calcification [43].

Generally, GGs present a slow-growing trend, and their clinical manifestations are slow. Surgical resection generally leads to good prognosis. In general, on the premise of preserving the integrity of neurologic function, a total tumor resection should be considered to be of the utmost priority. However, since the suprasellar region has a complex structure and is close to important blood vessels and nerves, biopsy has become an alternative method. According to the data, the 10-year survival rate of these patients after total tumor resection is 82–93% [44]. We believe that GG is also characterized by a low recurrence rate. According to the statistics of Professor Blümcke et al., among the 86 patients diagnosed with ganglioglioma (WHO grade I), only 1 patient relapsed after 7 years of follow-up, and no tumor recurrence was recorded in the 19 patients we counted [1]. Upon admission, patients can choose between craniotomy and transnasal endoscopic surgery, especially in the case of our study. The principal reason that we chose to perform transnasal endoscopic surgery was that it can better expose the important structure surrounding the tumor. Second, it is minimally invasive and scarless, and allows direct entry to the skull base. No manipulation of neurovascular structures is required. In addition, the transnasal approach allows us to obtain a good view of the hypothalamus and other structures. In comparison to traditional craniotomy, the incidence of brain injury and cerebrovascular accident is significantly lower, but cerebrospinal fluid leakage is more likely to occur [45]. For GGs that can be enhanced on preoperative MRI, some literature studies suggest that better results can be obtained with fluorescein-guided surgery, whether surgical resection or surgical biopsy [46]. Therefore, during the operation, we used artificial dura mater and a vascularized pedicle nasoseptal flap to repair the bottom structure of the sella, thus reduced the chance of cerebrospinal fluid leakage. We believe that endoscopic transnasal surgery is a better option. Currently, there is still no clear evidence supporting the need for adjuvant therapy following surgery for GGs. However, certain data show that adjuvant radiotherapy can prolong the interval of recurrence, although it does not improve the overall survival rate [47]. In addition, chemotherapy has also been applied for this type of tumor. In our literature review, Vajramani et al. [13] used temozolomide for chemotherapeutic purposes, and Spennato et al. [16] used temozolomide and imatinib. We cannot form a conclusion due to the limited results.

### 8. Conclusions

Ganglioglioma is a rare intracranial tumor occurring in adults and anywhere in the central nerve system; it occurs more often in the temporal lobe and is rarer in the suprasellar regions. A gross total surgical resection should be performed as first-choice treatment. Histopathological examinations play an important role in the precise diagnosis of these tumors. In the present study, we reported the first suprasellar ganglioglioma originating from the third ventricle floor, which was resected via transsphenoidal endoscopy and the patient showed good prognosis.

**Author Contributions:** S.L. (Shaoguang Li) wrote the manuscript. Y.X. wrote the manuscript and offered some information on the case. S.L. (Shaoguang Li) and Y.X. contributed equally to the manuscript. G.H., S.L. (Shigang Lv), P.S. and H.G. provided assistance in the writing of the manuscript and offered some information on the case. L.W. designed and supervised the study and wrote the manuscript. All authors have read and agreed to the published version of the manuscript.

**Funding:** This research was funded by the JiangXi Provincial Natural Science Foundation of China (No. 20192BAB205077) and the National Natural Science Foundation of China (No. 82160153).

**Institutional Review Board Statement:** The study was conducted according to the guidelines of the Declaration of Helsinki and approved by the Ethics Committee of The Second Affiliated Hospital of Nanchang University (protocol code "Review [2022] No. (008)" and date of approval "19 January 2022").

**Informed Consent Statement:** Written informed consent was obtained from the patient to publish this paper.

**Data Availability Statement:** Not applicable.

**Acknowledgments:** We sincerely appreciate the assistance and technical support offered by the Department of Neurosurgery, The Second Affiliated Hospital of Nanchang University.

**Conflicts of Interest:** The authors declare no conflict of interest.

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
