# Peer review of "Suprasellar Ganglioglioma Arising from the Third Ventricle Floor: A Case Report and Review of the Literature"

_tomography, doi:10.3390/tomography8060238_

Round 1
Reviewer 1 Report
I appreciated the work. The paper is easy to read and enjoyable, useful for the restricted number of surgeons performing this kind of surgery with an endonasal approach.
As a reader, I would read what is the natural history of the possible untouchable residuals: a remnant involving the stalk or encasing a vascular structure, will regrow and in which time-span? This is the core question to perform safe surgery on rare tumors of such an elusive behaviour. Add a paragraph concerning the regrowth/recurrence
Reviewer 2 Report
In this manuscript, the authors report the first case of a suprasellar ganglioglioma originating from the third ventricle floor, which was removed via transsphenoidal endoscopy. These are extremely rare tumors in general and even rarer when discovered in the suprasellar region. The authors provide the history of the case, a description of the surgical procedure documented with intraoperative images, the histopathological analysis of the tumor specimen, and a description of the postoperative course of the patient. Lastly, the authors provide a general review on ganglioglioma cases based on literature search. The manuscript is well written, and the case report is both detailed and interesting enough for the potential readers to be considered for publication. I only have a few comments for the authors which I hope they will kindly address.
1. The authors stated regarding the aspect of the margins of the operation area on a postoperative MRI that these were enhanced, but this enhancement was judged to be a sign of postoperative capillary reactivity. Based on this, the authors concluded that “All these results suggest that the tumor was completely removed” (line 114). However, a few lines down, the authors deem the operation to be a subtotal tumor resection (line 123). I think it would be beneficial for the potential reader if the authors could further clarify and correct these discrepancies.
2. Upon histopathological analysis, the tumor was deemed to contain low-grade astrocytic cells (i.e., the glial component) and to have a Ki67 index of about 1% (lines 91-92). Despite these findings, postoperative radiotherapy was still recommended to the patient about 1 month following surgery. While this course of action is not necessarily unusual, in light of the current guidelines, it would be interesting for the authors to further clarify why radiotherapy was still recommended in this case (i.e., low-grade astrocytic component and a low Ki67 index with a potentially total or near-total resection of the tumor). Was the opening of the third ventricle a potential factor behind this decision?
Other than the above, one minor comment is regarding a small typo that needs to be corrected: i.e., on line 63, the term “the arachnoid member” should read “the arachnoid membrane”. I conclude by thanking the authors for this interesting case.
